# Epidermal Growth Factor Receptor (EGFR) Pathway, Yes-Associated Protein (YAP) and the Regulation of Programmed Death-Ligand 1 (PD-L1) in Non-Small Cell Lung Cancer (NSCLC)

**DOI:** 10.3390/ijms20153821

**Published:** 2019-08-05

**Authors:** Ping-Chih Hsu, David M. Jablons, Cheng-Ta Yang, Liang You

**Affiliations:** 1Department of Surgery, Helen Diller Family Comprehensive Cancer Center, University of California, San Francisco, San Francisco, CA 94115, USA; 2Department of Thoracic Medicine, Chang Gung Memorial Hospital, Linkou, Taoyuan 33305, Taiwan; 3School of Medicine, Chang Gung University, Taoyuan 33302, Taiwan

**Keywords:** epidermal growth factor receptor (EGFR), yes-associated protein (YAP), programmed death-ligand 1 (PD-L1), non-small cell lung cancer (NSCLC), tyrosine kinase inhibitor (TKI), immune checkpoint inhibitors (ICIs)

## Abstract

The epidermal growth factor receptor (EGFR) pathway is a well-studied oncogenic pathway in human non-small cell lung cancer (NSCLC). A subset of advanced NSCLC patients (15–55%) have EGFR-driven mutations and benefit from treatment with EGFR-tyrosine kinase inhibitors (TKIs). Immune checkpoint inhibitors (ICIs) targeting the PD-1/PDL-1 axis are a new anti-cancer therapy for metastatic NSCLC. The anti-PD-1/PDL-1 ICIs showed promising efficacy (~30% response rate) and improved the survival of patients with metastatic NSCLC, but the role of anti-PD-1/PDL-1 ICIs for EGFR mutant NSCLC is not clear. YAP (yes-associated protein) is the main mediator of the Hippo pathway and has been identified as promoting cancer progression, drug resistance, and metastasis in NSCLC. Here, we review recent studies that examined the correlation between the EGFR, YAP pathways, and PD-L1 and demonstrate the mechanism by which EGFR and YAP regulate PD-L1 expression in human NSCLC. About 50% of EGFR mutant NSCLC patients acquire resistance to EGFR-TKIs without known targetable secondary mutations. Targeting YAP therapy is suggested as a potential treatment for NSCLC with acquired resistance to EGFR-TKIs. Future work should focus on the efficacy of YAP inhibitors in combination with immune checkpoint PD-L1/PD-1 blockade in EGFR mutant NSCLC without targetable resistant mutations.

## 1. The Epidermal Growth Factor Receptor (EGFR) Pathway in Non-Small Cell Lung Cancer (NSCLC)

EGFR is a transmembrane tyrosine kinase receptor, which is one of the ErbB family of receptors. The ErbB family includes EGFR (ErbB1), ERBB2 (HER2/neu), ERBB3 (HER3), and ERBB4 (HER4) [1,2]. The structure of EGFR is characterized by an extracellular ligand-binding domain, a transmembrane domain, and a cytoplasmic domain containing the tyrosine kinase region with tyrosine autophosphorylation sites. Earlier studies reported that EGFR overexpression appeared in 60% of NSCLC and is associated with poor prognosis [3,4]. The activating mutation in the EGFR kinase domain has been well studied and EGFR mutation is well-known as the main oncogenic driven mutation in some NSCLC [5,6,7,8]. 

The EGFR kinase domain is encoded by exons 18–24, which are clustered around the ATP-binding pocket of the enzyme. EGFR kinase domain mutations mainly occur at exon 18–21, and increase the kinase activity of EGFR, leading to the hyperactivation of downstream pro-survival signal pathways which promote tumorigenesis of NSCLC cells [8]. The main three downstream signaling pathways activated by EGFR are mitogen-activated protein kinases (MAPK)/extracellular signal-regulated kinases (ERK), phosphatidylinositol 3-kinase (PI3K)/Akt/mTOR, and interleukin 6(IL-6)/Janus kinase (JAK)/signal transducer and activator of transcription 3 (STAT3) signaling pathways [1,2,8]. 

The frequency of EGFR driver mutations in NSCLC differs according to ethnicity; rates range from 5% to 15% in Caucasians and from 40% to 55% in East Asians [9,10,11,12]. Two types of EGFR mutations, L858R (point mutations in exon 21 causing a leucine-to-arginine substitution at codon 858) and exon 19 deletion (in-frame deletions in exon 19), account for approximately 90% of all EGFR mutations [13,14,15,16]. NSCLC harboring EGFR L858R or exon 19 deletion mutations is sensitive to the EGFR-tyrosine kinase inhibitors (TKIs) including gefitinib, erlotinib, afatinib, dacomitinib, and osimertinib. Gefitinib and erlotinib are classified as 1st-generation EGFR-TKI; afatinib and dacomitinib are 2nd-generation, and osimertinib is 3rd-generation [13,14,15,16,17]. The 1st-generation EGFR-TKIs gefitinib and erlotinib have reversible binding to mutant EGFR, but are inactive to acquired T790M mutation [18,19,20,21,22]. The 2nd-generation EGFR-TKIs afatinib and dacomitinib have irreversible covalent binding to all ErbB receptors (EGFR, ErbB2, ErbB4, and ErbB heterodimers), and are inactive to T790M mutation [23,24,25,26]. The 3rd-generation EGFR-TKI osimertinib is characterized by irreversible covalent binding to mutant EGFR and activates on T790M mutation [27,28,29]. The efficacy of these EGFR-TKIs in treating advanced NSCLC harboring EGFR mutation had been investigated in several large prospective clinical trials (Table 1) [18,19,20,21,22,23,24,25,26,27,30]. These trials showed that the 1st and 2nd generation EGFR-TKIs yield response rates ranging from 60% to 80%, which were 2 times higher than that conventional chemotherapy in clinical trials [18,19,20,21,22,23,24,30]. They also yielded longer progression-free survival. Therefore, 1st and 2nd generation EGFR-TKIs have been widely used as first-line standard treatment for advanced NSCLC patients with sensitive EGFR mutations (L858R or exon 19 deletion). 

The T790M mutation is a point mutation in exon 20 with the substitution of methionine to threonine at amino acid position 790. T790M mutation is known as a “gate keeper” mutation in the kinase domain of EGFR, which alters the binding of 1st and 2nd generation EGFR-TKIs to the ATP-binding pocket [27,28,29]. T790M mutation accounts for 30–60% of secondary EGFR point mutations in NSCLC patients who developed acquired resistance to 1st and 2nd generation of EGFR-TKIs [27,28,29]. T790M mutation also appears in EGFR-TKI-naïve NSCLC patients, most of whose tumors do not respond to treatment with 1st and 2nd generation EGFR-TKIs [31]. Clinical trial evidence indicates that osimertinib, a 3rd generation EGFR-TKI developed to overcome the resistance of T790M, is effective (~70% response rate) in treating NSCLC patients who had acquired resistance to 1st and 2nd generation EGFR-TKIs because of the development of the T790M mutation [32,33]. In addition, a phase III clinical trial showed that osimertinib resulted in significant longer progression-free survival than gefitinib and erlotinib in untreated EGFR-mutated advanced NSCLC [27].

Although EGFR-TKIs are standard therapy for EGFR-mutated advanced NSCLC, studies to elucidate the mechanism of acquired resistance are ongoing. The development of new targeted therapy and rational combination strategies to overcome EGFR TKI resistance are being investigated in pre-clinical and clinical studies [34]. In addition, the role of therapies such as immune checkpoint inhibitors (ICIs) in combination with or sequentially following EGR-TKIs, remains to be explored. The oncogenic pathway of EGFR and the targeting EGFR mutations of EGFR-TKIs are summarized in Figure 1.

## 2. Programmed Cell Death Protein-1 (PD-1)/Programmed Death-Ligand 1 (PD-L1) Immune Checkpoint in NSCLC

Programmed cell death protein-1(PD-1; also known as CD279), an immune checkpoint receptor expressed on the cell surface of immune cells, including T-cell, B-cell, and myeloid cells, plays important roles in immune regulation [35,36,37]. PD-1 belongs to the CD28 family and delivers a negative signal when it interacts with its ligands—programmed death-ligand 1 (PD-L1) (also known as B7-H1 or CD274) and PD-L2 (known as B7-DC or CD273). PD-L1 is one of the ligands of PD-1 and is broadly expressed in healthy tissue cells including vascular endothelial cells, pancreatic islet cells, astrocytes, and corneal epithelial and endothelial cells. In normal human tissue cells, the binding of PD-1/PD-L1 promotes T-cell tolerance and escape from host immunity by downregulating CD8+ T-cell survival and effector function. The dysregulation and deficiency of PD-1 can cause autoimmune diseases [35,36,37]. When PD-L1 is expressed in cancer cells, it can engage the immune checkpoint PD-1/PD-L1 axis to escape antitumor immune responses to prevent the immune system from killing cancer cells [37,38,39]. Therefore, immune checkpoint inhibitors (ICIs) targeting the PD-1/PDL-1 axis were developed as an anti-cancer therapy [37,38,39,40,41,42,43]. 

Immunotherapies targeting the PD-1/PDL-1 axis were studied in clinical trials worldwide, and used to treat various solid and hematological malignant tumors including NSCLC [37,38]. PD-L1 has been shown to be expressed in human NSCLC, and anti-PD-1/PD-L1 ICIs have been used to treat advanced NSCLC [43,44,45,46]. The immunochemistry staining of cell-surface PD-L1 expression level was used as a predictive biomarker for anti-PD-1/PD-L1 ICIs in treating NSCLC [47,48]. To date, 4 anti-PD-1/PD-L1 ICIs, pembrolizumab, nivolumab (anti-PD-1), atezolizumab and durvalumab (anti-PD-L1), have been approved by the U.S. Food and Drug Administration (FDA) for the treatment of advanced NSCLC. The efficacy of all 4 was shown in clinical trials (Table 2) [48,49,50,51,52,53,54,55]. Pembrolizumab, nivolumab, and atezolizumab improved overall survival significantly when compared to conventional chemotherapy in treating metastatic NSCLC [48,49,50,51,52,53]. Pembrolizumab is the only anti-PD-1/PD-L1 inhibitor authorized in first line monotherapy for advanced non-squamous NSCLC without driver gene mutations (EGFR mutation and ALK fusion were excluded) expressing PD-L1 ≥ 50% of tumor cells and in second line monotherapy for advanced non-squamous or squamous NSCLC expressing PD-L1 ≥ 1%. Nivolumab and atezolizumab are authorized for tumor PD-L1 expression-negative and -positive advanced non-squamous or squamous NSCLC which were previously treated by at least one-line of chemotherapy. Durvalumab yielded significantly longer progression-free survival and overall survival than placebo in stage III NSCLC after chemoradiotherapy [54,55]. Therefore, durvalumab currently is authorized for unresectable stage III NSCLC after chemoradiotherapy despite the level of tumor PD-L1 expression condition. 

Patients with EGFR-mutant advanced NSCLC account for just 5–14% of patients in the clinical trials of the four anti-PD-1/PD-L1 ICIs [48,50,51,52,53,54,55]. In addition, NSCLC patients with sensitive EGFR mutations were excluded in the clinical trial of KEYNOTE-024, which compared the efficacy of pembrolizumab and chemotherapy for PD-L1–positive NSCLC [49]. In these clinical trials, the EGFR-mutant NSCLC patients were previously treated with EGFR-TKIs and at least one chemotherapy regimen. Therefore, the role of anti-PD-1/PD-L1 ICIs in EGFR mutant NSCLC is currently not clear.

Two other anti-PD-1/PD-L1 ICIs—cemiplimab (anti-PD-1) and avelumab (anti-PD-L1)—are not yet approved for NSCLC in the U.S. In a recent study, cemiplimab led to an approximately 50% response rate for advanced cutaneous squamous-cell carcinoma, and was approved for this purpose by the FDA in September 2018 [56]. A phase II clinical trial (NCT03430063) to investigate the efficacy of cemiplimab in advanced NSCLC is ongoing. The results of a phase II clinical trial (NCT02155647; JAVELIN Merkel 200) showed that avelumab had a response rate of 33% for the treatment of advanced Merkel-cell carcinoma, an aggressive type of skin cancer [57]. Avelumab was FDA-approved in March 2017. The efficacy of avelumab for the treatment of advanced NSCLC is under investigation in several clinical trials [58,59].

The rationale for PD-1/PD-L1 ICIs in NSCLC is summarized in Figure 2.

## 3. Yes-Associated Protein (YAP) in Human NSCLC

YAP (yes-associated protein) is the main mediator of the Hippo (also known as the Salvador–Warts–Hippo) signaling pathway and negatively regulated by upstream components of the Hippo pathway. When the Hippo pathway is activated, YAP will be sequestered by Hippo kinase in the cytoplasm and degraded. Conversely, when the Hippo pathway is inactivated, YAP will translocate into the nucleus to form the complexes with transcriptional enhancer factors (TEF; also known as TEAD). The combination of YAP and TEAD in the nucleus activates transcription of downstream genes. In normal tissues, the Hippo/YAP pathway plays an important role in regulating organ size [60,61,62,63,64]. Elevated levels and nuclear localization of YAP have been found in various cancers as a result of loss of Hippo signaling by mutation, and downregulation of the core Hippo components. YAP has the ability to promote stem cell renewal and differentiation, a crucial step in oncogenic transformation [58], and reportedly promotes cancer development in various human cancers including lung, breast, liver, skin, colon, and ovarian cancer [60,61,62,63,64]. 

In physiological conditions, YAP is required for mammalian lung development and contributes to repair and regeneration after lung injury. Previous studies showed that a balance between nuclear and cytoplasmic YAP is critical in regulating proximal-distal patterning of the lung development in mouse models [65,66]. YAP has been identified in human NSCLC, and overexpression of YAP in NSCLC is associated with cancer progression and poor prognosis [67,68]. Several previous studies reported that YAP also promotes cancer invasion, drug resistance, and metastasis in NSCLC [69,70,71,72,73,74,75]. Inhibition of YAP increases cytotoxicity of chemotherapy and sensitivity to various target therapies including EGFR-TKIs, BRAF, and MEK inhibitors in NSCLC [70,71,72,73]. Increased YAP expression in tumors harboring BRAF V600E mutation is a biomarker of worse initial response to RAF and MEK inhibitors in patients [70]. The combination of YAP and RAF or MEK inhibition synergizes the cytotoxicity of RAF and MEK inhibitors to BRAF- and K-ras-mutant NSCLC cells [70]. Few studies have investigated the role of YAP in promoting NSCLC metastasis in vivo. One study found that increased YAP expression was correlated with risk of metastasis and poor prognosis in clinical samples from NSCLC patients [67]. Dubois et al. found that the tumor suppressor gene RASSF1A suppresses epithelial-to-mesenchymal transition (EMT) ability of human NSCLC cells by inhibiting YAP through the GEF-H1/RhoB pathway. They demonstrated that the effect of inhibiting YAP by the RASSF1A gene suppresses NSCLC metastasis in a mouse model [74]. Another recent study found that NSCLC cell lines (H2030-BrM3 and PC9-BrM3) [75,76] with high potential for brain metastasis had increased YAP expression when compared to their parental cell lines (H2030 and PC9), and indicated that YAP plays an important role in promoting NSCLC brain metastasis [77]. This study also demonstrated that inhibition of YAP decreases expression of its downstream genes connective tissue growth factor (CTGF) and cysteine-rich angiogenic inducer 61 (CYR61), as well as migration and invasion abilities in H2030-BrM3 cells. In addition, direct inhibition of YAP by short hairpin RNA suppresses brain metastasis of NSCLC H2030-BrM3 cell line in a mouse model [77]. The two previous studies showed that the inhibition of YAP suppresses human NSCLC metastasis ability in vivo. Taken together, the findings of these studies provide perspectives for further development and investigation of drugs targeting YAP for altering the drug resistance in NSCLC. Further studies to investigate YAP as a therapeutic target for metastatic NSCLC without targetable driven mutation is also warranted. 

## 4. The Interaction between EGFR and YAP Signaling Pathways in NSCLC

The EGFR and its downstream signaling Src/Ras/Raf/MEK/ERK pathway are highly associated with human NSCLC [78,79,80]. The EGFR/MAPK/ERK signaling pathway has crosstalk with the Hippo/YAP signaling pathway and is involved in the positive regulation of YAP oncogenic function in various cancers [81,82,83,84,85]. In one study [82], YAP was significantly associated with ERK 1/2 expression in clinical lung adenocarcinoma tissues by immunohistochemistry staining, and inhibition of ERK 1/2 downregulated YAP protein level expression through accelerating YAP protein degradation in human NSCLC cells. The same study also found that forced overexpression of the ERK2 gene rescued YAP protein expression during ERK 1/2 depletion [82]. Another study demonstrated that forced overexpression of YAP promotes resistance to EGFR-TKI erlotinib in EGFR mutant NSCLC cells, and that knock down of YAP increases the cytotoxicity of erlotinib to NSCLC cell line H1975 (L858R + T790M mutations) [86]. Several studies found that activation of YAP enhances the downstream gene expression of EGFR ligands such as Amphiregulin (AREG) and Neuregulin 1 (NRG-1), and ERBB3 and ERBB4, to form an autocrine loop and reinforce the MAPK signaling pathway to induce cancer progression and drug resistance [84,85,86,87,88,89]. The EGFR/MAPK signaling pathway was reported to regulate YAP through inhibiting Hippo kinases that phosphorylate and then degrade YAP protein in cytoplasm [86]. For instance, MAPK can regulate Ajuba family LIM domain protein (Jub), Wilms tumor protein 1-interacting protein (WTIP), or KIBRA, which are involved in the regulation of the Hippo pathway [90,91]. Raf-1 and oncogenic protein of MAPK was also shown to regulate hippo kinase mammalian sterile-20 like kinase 2 (MST2) [92].

K-ras is an oncogenic protein of the MAPK signaling pathway, and its gene mutation frequently occurs in NSCLC patients (15–30%). There is still no approved effective target therapy for K-ras mutant metastatic NSCLC [93]. According to one study [94], some K-ras mutant NSCLC cell lines, including A549, H23, and SK-LU-1, were not K-ras-dependent cells and do not require K-ras to maintain viability, and inhibiting or knocking down K-ras does not suppress the proliferation, migration, and invasion abilities in these cells. Two studies [95,96] reported that YAP appears to take over K-ras as a cancer driver in K-ras mutant NSCLC and found that some K-ras mutant NSCLC cells relapsed after K-ras extinction without re-expression of the K-ras transgene. Amplification of the YAP gene was found in the relapsed tumors without re-expression of K-ras, and knockdown of YAP affected cancer cell growth in vivo. In addition, YAP interacts with FOS to activate MAPK signaling to induce an epithelial-mesenchymal transition (EMT) program in the absence of K-ras signaling in lung cancer [95,96,97]. Together, the findings of these studies suggest that YAP is a compensated central driver for K-ras-dependent NSCLC when there is loss of K-ras signaling [95,96,97]. The EGFR/MAPK signaling pathway interacts with YAP positively to promote cancer progression, drug resistance, and metastasis in human NSCLC. Figure 3 summarizes the mechanism by which EGFR and YAP regulate PD-L1 expression in human NSCLC, according to the findings of previous studies.

## 5. The EGFR and YAP Signaling Pathways Regulate PD-L1 Expression in NSCLC

Though PD-L1/PD-1 ICIs are suggested for metastatic NSCLC patients whose tumors strongly express PD-L1 (tumor proportion scores ≧50%), more than 50% of tumors with strong PD-L1 expression do not respond to PD-1/PD-L1 inhibitors [47,49]. The mechanism of how tumor PD-L1 expression is regulated by oncogenic signaling pathways is in the early stages of being investigated. The goal of these studies is to help identify biomarkers and develop therapeutic strategies for clinical use of anti-PD-1/PD-L1 ICIs. 

The interaction between EGFR and cancer immunity in the tumor microenvironment has been established. Myeloid-derived suppressor cells (MDSCs) secrete immunosuppressive factors COX2, PGE2, ARG1, and IL-6 in the tumor microenvironment to reduce the cytotoxicity of anti-EGFR target therapy in pancreatic cancer [98]. Inhibition of MDSCs with downregulating the downstream cytokines restores the cytotoxicity of anti-EGFR therapy [98]. Patients with advanced EGFR mutant NSCLC and increased blood S100A9+ MDSCs have had a poor treatment response to EGFR-TKIs [99]. The MDSCs were also involved in the regulation of PD-L1 expression in NSCLC [100,101], and reduction of MDSCs by radiation therapy enhances the treatment response of anti-PD-1/PD-L1 ICIs in NSCLC [100,101]. Taken together, these studies suggest that cancer immunity in the tumor microenvironment of NSCLC not only alters the efficacy of EGFR-TKIs but also the anti-PD-1/PD-L1 ICIs. 

Little is known about the role of EGFR in regulating PD-L1 expression in human NSCLC. Two recent studies that investigated the role of EGFR in regulating PD-L1 expression in human EGFR-driven NSCLC found that the activation of EGFR through epidermal growth factor stimulation and mutations of 19 deletion or L858R induced PD-L1 expression in human NSCLC cells [102,103]. The studies showed that inhibiting EGFR by EGFR-TKI downregulated PD-L1 expression in human EGFR-driven NSCLC cells. One of the studies concluded that EGFR activation upregulates PD-L1 expression through the p-ERK1/2/p-c-Jun signaling pathway, not through the p-AKT/p-S6 signaling pathway, and demonstrated that inhibiting EGFR by EGFR-TKIs reversed the apoptosis of T cells and enhanced the production of interferon (IFN)-γ [102]. The other study concluded that EGFR regulates PD-L1 expression and cell proliferation via the IL-6/JAK/STAT3 signaling pathway in EGFR mutant NSCLC [103]. Another study showed that inhibition of cytotoxic T lymphocyte antigen 4 (CTLA4) induced PD-L1 expression through activation the EGFR/ERK pathway in human NSCLC [104]. That study further demonstrated that inhibiting EGFR by genomic knockout or EGFR-TKI treatment suppressed the anti-CTLA4 antibody induced PD-L1 expression in NSCLC cells [104]. The findings of this study partly explained why single anti-CTLA-4 inhibitor treatment is not as effective as anti-PD-1/PD-L1 ICIs for advanced NSCLC clinically, either in objective response rate or prolonging survival. Together, these studies indicate that the EGFR signaling pathway plays an important role in regulating PD-L1 expression in human NSCLC cells harboring EGFR mutation [102,103,104].

The role of YAP in cancer immunity has just begun to be explored. YAP was reported to be a negative regulator of innate immunity because of its interaction with interferon regulatory factor 3 [105,106]. Recent studies found that YAP regulates tumor-associated immune cells in the tumor microenvironment, including MDSCs, tumor-associated macrophages, and regulatory T cells [107,108,109]. These studies demonstrated that chemical YAP antagonism and knockout or blockade of YAP improves antitumor immunity in mouse models, which suggests that YAP is involved in regulating the immune checkpoint [107,108,109]. 

We recently investigated YAP’s role in the regulation of PD-L1 expression in human NSCLC [110]. In human NSCLC tumor samples, immunohistochemistry showed that positive nuclear YAP staining was significantly associated with positive PD-L1 expression. We demonstrated that inhibition of YAP by genomic knockdown with small interfering RNAs (siRNAs) and the chemical compound verteporfin decreased the protein and mRNA levels of PD-L1 in NSCLC cells. Another recent study showed that forced overexpression of the YAP gene increased the PD-L1 protein expression level in NSCLC A549 cells, which have low YAP and PD-L1 expression [111]. In addition, we found that the precipitation of the PD-L1 enhancer region encompasses two putative TEAD binding sites in a chromatin immunoprecipitation (ChIP) assay by using a YAP-specific monoclonal antibody [110,111]. A recent study that showed YAP regulates PD-L1 expression in EGFR-TKI-resistant NSCLC, that gefitinib-resistant PC9 cells (exon 19 deletion mutation) have increased YAP and PD-L1 protein expression when compared to parental PC9 cells, and that YAP knockdown decreased the expression of PD-L1 in gefitinib-resistant PC9 cells [112]. The findings of these studies indicate that YAP regulates the transcription and expression of PD-L1 in human NSCLC [110,111,112,113].

Other recent work has shown that Hippo kinases including mammalian STE20-like kinase 1 and 2 (MST1/2), and large tumor suppressor 1 and 2 (LATS1/2), suppress PD-L1 expression, whereas the two main mediators of the Hippo pathway—TAZ and YAP—enhance PD-L1 levels in lung cancer cell lines [114,115]. YAP was identified as a critical mediator for c-Jun in several studies [116,117,118,119]. Collectively, the findings of these studies indicate that EGFR regulate PD-L1 expression at least partly through the YAP signaling pathway in human NSCLC. Figure 3 summarizes the mechanism by which EGFR does this.

## 6. Future Perspectives 

### 6.1. The Role of Anti-PD-1/PD-L1 ICIs in EGFR Mutant NSCLC

Although the anti-PD-1/PD-L1 ICIs have shown promising results, with ~30% of NSCLC responding in clinical trials, the role of these ICIs in the treatment for advanced EGFR mutant NSCLC is not clear yet. Currently, EGFR-TKIs yield very high efficacy in treating advanced EGFR mutant NSCLC (60–80% response rate and more than 10 months of progression-free survival) [18,19,20,21,22,23,24,25,26,27,30]. However, anti-PD-1/PD-L1 ICIs did not improve survival when compared with conventional chemotherapy in patients with advanced EGFR-mutant NSCLC [120]. A recent phase 1/2 clinical trial (NCT02039674; KEYNOTE-021) investigated the feasibility of combining erlotinib or gefitinib with pembrolizumab (anti-PD-1 inhibitor) as first-line therapy for advanced NSCLC with sensitive EGFR mutation. This trial showed that pembrolizumab in combination with erlotinib or gefitinib did not improve objective response rate compared with erlotinib or gefitinib monotherapy. This study failed to show synergistic tumor-cell killing effects in EGFR mutant NSCLC by the combination treatment consisting of EGFR-TKIs and anti-PD-1 antibody [121]. The efficacy of anti-PD-1/PD-L1 immunotherapy in combination with EGFR-TKIs for advanced EGFR mutant NSCLC is unclear and is being investigated in several early phase clinical trials [122]. For advanced NSCLC with acquired resistance to 1st or 2nd generation EGFR-TKI therapy, the 3rd generation EGFR-TKI osimertinib is suggested as therapy prior to anti-PD-1/PD-L1 ICIs if T790M mutation is detected [27,33,123]. However, more than 50% of patients with EGFR-mutant NSCLC have acquired resistance to 1st, 2rd, and 3rd generation EGFR-TKIs without known targetable secondary mutations [34,124,125,126]. Currently, cytotoxic chemotherapy is the main rescue treatment for patients with advanced EGFR-mutant NSCLC who do not have known, targetable, acquired resistant mutations [15,126,127,128]. 

Adaptive immune resistance is one resistant mechanism through which cancer changes its phenotype in response to cytotoxic target therapy and evades the therapy [129]. A recent clinical trial (ATLANTIC) showed responses to anti-PD-1/PD-L1 ICIs in some patients with EGFR mutant advanced NSCLC pre-treated by EGFR-TKIs and chemotherapy [130]. An earlier study showed that in EGFR mutant NSCLC with resistance to EGFR-TKIs, T790M-negative patients had a higher positive PD-L1 expression rate than T790M-positive patients and were more likely to benefit from nivolumab (anti-PD-1) therapy after EGFR-TKI treatment [131]. 

Anti-PD-1/PD-L1 ICIs in combination with chemotherapy may be more effective than anti-PD-1/PD-L1 ICIs alone for the treatment of advanced NSCLC [132]. Therefore, this combination therapy may be suggested as sequential treatment for patients with EGFR-mutant NSCLC who do not have known targetable acquired resistant mutations, but efficacy studies are needed. An ongoing clinical trial (NCT02864251; CheckMate722) will assess the role of ICIs (nivolumab + chemotherapy or nivolumab + ipilimumab versus chemotherapy) in the treatment of T790M mutation negative NSCLC patients who have acquired resistance to EGFR-TKI. Another ongoing trial (NCT03515837; KEYNOTE-789) will assess the efficacy of pembrolizumab in combination with chemotherapy (chemotherapy with or without pembrolizumab). Unlike the CheckMate722 trial, the KEYNOTE-789 trial will allow recruitment of T790M mutant patients who have acquired resistance to osimertinib treatment. 

### 6.2. Targeting YAP Thrapy in Combination with ICIs in EGFR-TKI Resistant NSCLC 

Therapies targeting YAP have been reported to be potential treatments to overcome EGFR-TKI resistance in human NSCLC [70,83,84,86,133]. A recent study reported that EGFR-TKI treatment alters the tumor microenvironment in EGFR mutant NSCLC and found the level of mononuclear MDSCs was consistently elevated for the duration of EGFR-TKI therapy. The increase in serum inflammatory factors IL-10 and CCL-2 after EGFR-TKI treatment was also found in vivo [134]. The findings of increased MDSC and inflammatory factors associated with EGFR-TKI treatment may partly explain why most EGFR-TKI-resistant NSCLC patients have resistance to anti-PD-L1/PD-1 ICIs. The treatments and strategies that modulate the tumor microenvironment are thought to improve antitumor immune responses [134]. Therefore, targeting YAP therapies may potentially enhance the efficacy anti-PD-L1/PD-1 ICIs for advanced NSCLC because YAP plays critical role in regulating tumor immunity and PD-L1 expression. [107,108,109,110,111,112]. Small molecule inhibitors or drugs have been discovered that inhibit YAP, including dasatinib, statins, A35, JQ1, norcantharidin, agave, MLN8237, and dobutamine [135]. More in vitro and in vivo experiments are warranted to investigate the efficacy of YAP inhibition in combination with anti-PD-L1/PD-1 ICIs for EGFR-TKI resistant NSCLC. 

An alternative strategy to inhibit YAP can focus on certain oncogenic pathways or mediators. For instance, inhibition of MEK1/2 or ERK1/2 accelerates YAP degradation in human NSCLC [82]. A recent study showed that the combination of MEK inhibitor anti-PD-1/PD-L1 antibodies synergistically increased antitumor response and prolonged survival outcome in a mouse model [136]. Two small molecule compounds, verteporfin and protoporphyrin IX, inhibit YAP-TEAD complex formation [61,137]. Cyclin-dependent kinase 1 (CDK1) has been shown to phosphorylate YAP at multiple sites during the G2/M phase of the cell cycle in cancer cells [138,139]. Cyclin-dependent kinase 9 (CDK9) was shown to be a key component of transcription mediators and the elongation complex in promoting transcriptional activation driven by the YAP/TEAD complex. Therefore, CDK1 and CDK9 inhibitors are potential drugs that target YAP. CDK9 inhibitors including flavopiridol, dinaciclib, seliciclib, SNS-032, and RGB-286638 have been evaluated in pre-clinal and clinical studies [140,141,142]. A recent study showed that inhibition of CDK9 sensitizes to the anti-PD-1 immune checkpoint inhibitor in vivo [143]. Future work to investigate the efficacy of YAP inhibitors in combination with anti-PD-1/PD-L-1 ICIs in NSCLC patients with acquired resistance to EGFR-TKIs is feasible. Figure 4 summarizes the role of YAP in NSCLC with EGFR-independent resistance to EGFR-TKIs and the strategies of YAP blockade in combination with anti-PD-1/PD-L1 ICIs.

## 7. Conclusions 

Our review indicates that the EGFR and YAP pathways regulate the tumor immune checkpoint PD-1/PD-L1 in EGFR-mutant NSCLC, and that EGFR-TKIs are appropriate therapy prior to ICIs when sensitive mutations appear. Future work should focus on the efficacy of YAP inhibitors in combination with immune checkpoint PD-L1/PD-1 blockade in EGFR mutant NSCLC without targetable resistant mutations.

## Figures and Tables

**Figure 1 ijms-20-03821-f001:**
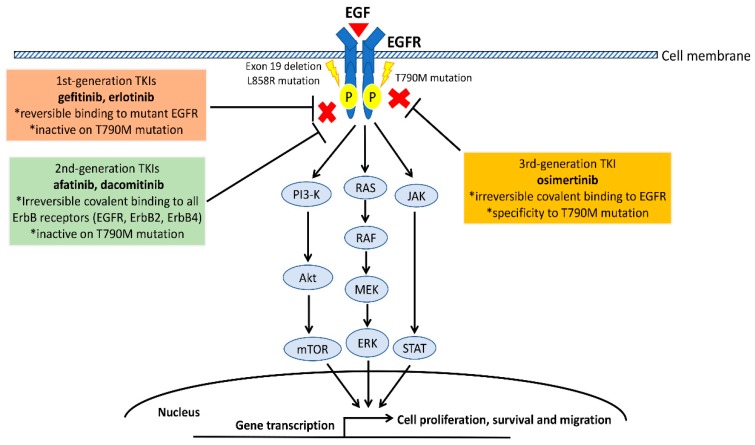
The Epidermal growth factor receptor (EGFR) pathway in non-small cell lung cancer (NSCLC). EGFR kinase domain mutations including exon 19 deletion, L858R and T790M increase kinase activity of EGFR, leading to the hyperactivation of downstream signaling pathways including MAPK, PI3K/Akt/mTOR, and IL-6/JAK/STAT3 which promote tumorigenesis of NSCLC cells. The three generations of EGFR-TKIs differ with respect to how they bind to different EGFR mutations and which EGFR mutations are active or inactive.

**Figure 2 ijms-20-03821-f002:**
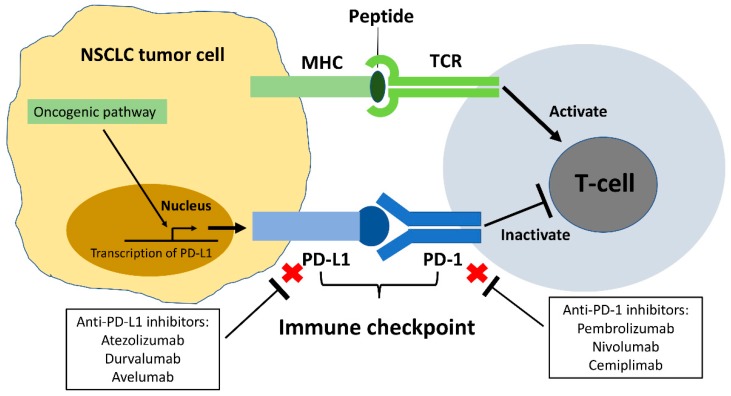
PD-1/PD-L1 immune checkpoint in NSCLC. In NSCLC cells, the binding of PD-1 and PD-L1 promotes T-cell tolerance and escape from host immunity. Immunotherapy targeting the PD-1/PD-L1 immune checkpoints has used to treat metastatic NSCLC. Pembrolizumab, nivolumab and cemiplimab are anti-PD-1 inhibitors, and atezolizumab, durvalumab and avelumab are anti-PD-L1 inhibitors.

**Figure 3 ijms-20-03821-f003:**
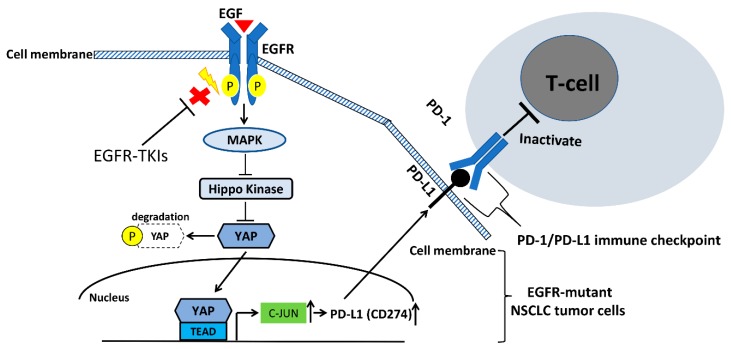
The EGFR pathway regulates PD-L1 expression in NSCLC. Stimulation of EGF or activation of EGFR kinase domain induced PD-L1 expression may be through the activating Hippo/yes-associated protein (YAP) signaling pathway in human NSCLC cells. Inhibition of EGFR by EGFR-TKIs downregulates PD-L1 expression in human EGFR-driven NSCLC cells.

**Figure 4 ijms-20-03821-f004:**
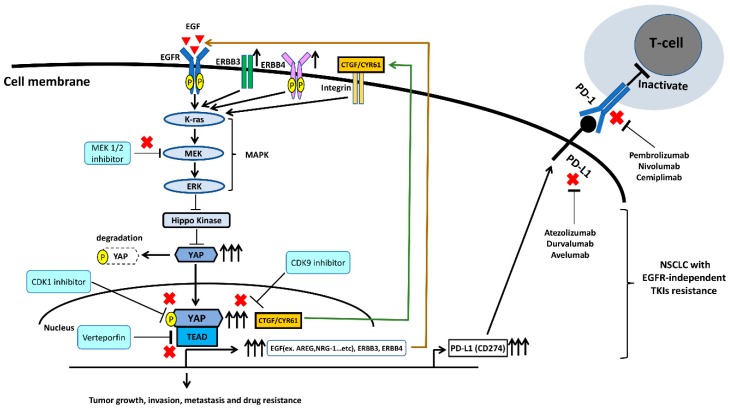
In EGFR mutant NSCLC without targetable resistant mutations, activation of YAP enhances the downstream genes expression of EGFs, ERBB3, ERBB4, CTGF, and CYR61 to form an autocrine loop and reinforce MAPK signaling pathway to induce cancer progression, drugs resistance, and immune escape. YAP blockade by MEK 1/2, CDK1, CDK9, and YAP-TEAD complex inhibitors in combination with anti-PD-1/PD-L1 ICIs may be a future therapeutic strategy for the treatment of EGFR mutant NSCLC without targetable resistant mutations.

**Table 1 ijms-20-03821-t001:** Summary of trials of first-line therapy using three generations of Epidermal Growth Factor Receptor (EGFR)-tyrosine kinase inhibitors (TKIs) in patients with mutation-positive non-small cell lung cancer (NSCLC).

Trial name	Treatment Arms	N	Mutation	Response Rate (%)	Median PFS, Months (HR [95%CI])	Median OS, Months (HR [95%CI])
IPASS [18]	Gefitinib vs. Cp/Pac	132 vs. 129	Exon 18–21	71.2 vs. 7.3 (*p* < 0.001)	9.5 vs. 6.3 (0.48 [0.36–0.64]; *p* < 0.001)	21.6 vs. 21.9 (1.00 [0.76–1.33]; *p* = 0.990)
WJTOG3405 [19]	Gefitinib vs. Cis/Doc	86 vs. 86	Exon 19 and 21	62.1 vs. 32.2 (*p* < 0.0001)	9.6 vs. 6.6 (0.52 [0.38–0.72]; *p* < 0.001)	35.5 vs. 38.8 (1.19 [0.77–1.83]; *p* = 0.443)
NEJ002 [20]	Gefitinib vs. Cp/Pac	114 vs. 114	Exon 18–21	73.7 vs. 30.7 (*p* < 0.001)	10.8 vs. 5.4 (0.32 [0.24–0.44]; *p* < 0.001)	27.7 vs. 26.6 (0.89 [0.63–1.24]; *p* = 0.483)
OPTIMAL [21]	Erlotinib vs. Cp/Gem	82 vs. 72	Exon 19 and 21	83 vs. 36 (*p* < 0.0001)	13.7 vs. 4.6 (0.16 [0.10–0.26]; *p* < 0.0001)	22.7 vs. 28.9 (1.04 [0.69–1.58]; *p* = 0.692)
EURTAC [22]	Erlotinib vs. Cis/Gem or Cp/Doc	86 vs. 87	Exon 19 and 21	64 vs. 18 (*p* < 0.0001)	10.4 vs. 5.1 (0.34 [0.23–0.49]; *p* < 0.0001)	22.9 vs. 20.8 (0.93 [0.64–1.36]; *p* = 0.71)
LUX-Lung 3 [24,30]	Afatinib vs. Cis/Pem	230 vs. 115	Exon 18–21	56 vs. 23 (*p* = 0.001)	11.1 vs. 6.9 (0.58 [0.43–0.78]; *p* < 0.0001)	28.2 vs. 28.2 (0.88 [0.66–1.17]; *p* = 0.39)
LUX-Lung 6 [23,24]	Afatinib vs. Cis/Gem	242 vs. 122	Exon 18–21	67 vs. 23(*p* < 0.0001)	11.0 vs. 5.6 (0.28 [0.20–0.39]; *p* < 0.0001)	23.1 vs. 23.5 (0.93 [0.72–1.22]; *p* = 0.61)
ARCHER1050 [25,26]	Dacomitinib vs. Gefitinib	227 vs. 225	Exon 19 and 21	74.9 vs. 71.6 (*p* = 0.4234)	14.7 vs. 9.2 (0·59[0.47–0.74]; *p* < 0·0001)	34.1 vs. 26.8 (0.760 [0.58–0.99]; *p* = 0.044)
FLAURA [27]	Osimertinib vs. Gefitinib or Erlotinib	279 vs. 277	Exon 19 and 21	80 vs76 (*p* = 0.24)	18.9 vs.10.2 (0.46 [0.37–0.57]; *p* < 0.001)	Not reached

Cp: Carboplatin; Pac: Paclitaxel; CI: Confidence interval; Cis: Cisplatin; Doc: Docetaxel; Gem: Gemcitabine; HR: hazard ratio; OS: Overall survival; Pem: Pemetrexed; PFS: Progression-free survival.

**Table 2 ijms-20-03821-t002:** Summary of anti-PD-1/PD-L1 immune checkpoint inhibitor (ICI) monotherapy in clinical trials.

Trial Name	PD-1 or PD-L1 Inhibitors	NSCLC Population	EGFR Mutation Rate (%)	Response Rate of ICIs (%)	Median OS, Months in ICI Treatment Group (HR [95%CI])
KEYNOTE-010 [43]	Pembrolizumab 2 mg/kg and 10 mg/kg groups	PD-L1 positive (≥1% expression)	6%–9%	19% (2 mg/kg) 19% (10 mg/kg)	10.4 (2 mg/kg) (0.71[0.58–0.88]; *p* = 0.0008) 12.7 (10 mg/kg) (0.61[0.49–0.75]; *p* < 0.0001)
KEYNOTE-024 [46]	Pembrolizumab 200mg	PD-L1 positive (≥50% expression); EGFR-wild-type ALK-wild-type	NA	44.8%	30.0 (0.49[0.34 to 0.69]; *p* = 0.002)
CheckMate 017 [47]	Nivolumab 3 mg/kg	Squamous NSCLC	NA	20%	9.2 (0.59 [0.44–0.79]; *p* < 0.001)
CheckMate 057 [48]	Nivolumab 3 mg/kg	Non-squamous NSCLC	14%	19%	12.2 (0.73[0.59–0.89]; *p* = 0.002)
OAK [50]	Atezolizumab 1200 mg	PD-L1 positive and negative expression enrolled	10%	14%	13.8 (0.73[0.62–0.87]; *p* = 0.0003)
PACIFIC [51]	Durvalumab 10 mg/kg	Stage III after chemoradiotherapy	6%	28.4	Not reached 0.68 [0.47–0.997]; *p* = 0.0025

CI: confidence interval; HR: hazard ratio; NA: not applicable; OS: overall survival; PFS: progression-free survival.

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
