# Peer review of "Epidermal Growth Factor Receptor (EGFR) Pathway, Yes-Associated Protein (YAP) and the Regulation of Programmed Death-Ligand 1 (PD-L1) in Non-Small Cell Lung Cancer (NSCLC)"

_ijms, 2019, doi:10.3390/ijms20153821_

Round 1

Reviewer 1 Report

The manuscript by Ping-Chih Hsu et al. is a review article about the correlation of the epidermal growth factor receptor (EGFR) pathway, the Yes-associated protein (YAP) pathway and the Programmed death-ligand 1 (PD-L1) expression with focus on the treatment of human non-small cell lung cancer. 

The article is very rich in information and well structured. Various treatment regimes are discussed with explicit response rates tabulated in a transparent way. The different pathways are well illustrated. The article reflects the relation between response rate to particular treatments and mutation status of pathway related proteins. 

As conclusion the authors make a specific recommendation for further studies, of the drug class that should be tested on a specific group of patients. 

The biology of tumours is often very complex and heterogeneous. But seeing the necessity of generating guidance for treatment the authors do their best to reach out to such a conclusion accepting the limited understanding of each individual case. 

Author Response

Point 1: English language and style are fine/minor spell check required ʉ۬

Response 1: The manuscript has been revised by our department’s scientific editor.

Point 2: The manuscript by Ping-Chih Hsu et al. is a review article about the correlation of the epidermal growth factor receptor (EGFR) pathway, the Yes-associated protein (YAP) pathway and the Programmed death-ligand 1 (PD-L1) expression with focus on the treatment of human non-small cell lung cancer. 

The article is very rich in information and well structured. Various treatment regimes are discussed with explicit response rates tabulated in a transparent way. The different pathways are well illustrated. The article reflects the relation between response rate to particular treatments and mutation status of pathway related proteins. As conclusion the authors make a specific recommendation for further studies, of the drug class that should be tested on a specific group of patients. 

The biology of tumours is often very complex and heterogeneous. But seeing the necessity of generating guidance for treatment the authors do their best to reach out to such a conclusion accepting the limited understanding of each individual case. 

Response 2:  We appreciate the reviewer’s enthusiastic comments about our review.

Reviewer 2 Report

In the review article “Epidermal Growth Factor Receptor (EGFR) Pathway, Yes-Associated Protein (YAP) and the regulation of Programmed death-ligand 1 (PD-L1) in Non-Small Cell Lung cancer (NSCLC)” by Hsu and colleagues, the authors review evidence that there is a link between EGFR expression, the Hippo signalling pathway and PD-L1 expression on tumour cells.

The authors describe recent clinical findings with great clarity and make a convincing point of their hypothesis. I read their manuscript with great pleasure and discovered several aspects I had so far not been aware of.

Nevertheless, there are still a few minor aspects that should be corrected/improved:

i)         It would be helpful to explain, how the different generations of EGFR inhibitors differ from each other (reversible, irreversible & targeting exclusively T790M mutants)

ii)        Figure 4 suggests that EGF could bind to ERBB3 and ERBB4 – which is of course incorrect. This aspect should be corrected.

iii)       An aspect worthwhile discussing in this context is the role Immune Check-Point Inhibitors play in reinvigorating tumor-specific immune responses and the effects EGFR inhibitors have on the immune system.

Author Response

In the review article “Epidermal Growth Factor Receptor (EGFR) Pathway, Yes-Associated Protein (YAP) and the regulation of Programmed death-ligand 1 (PD-L1) in Non-Small Cell Lung cancer (NSCLC)” by Hsu and colleagues, the authors review evidence that there is a link between EGFR expression, the Hippo signalling pathway and PD-L1 expression on tumour cells.

The authors describe recent clinical findings with great clarity and make a convincing point of their hypothesis. I read their manuscript with great pleasure and discovered several aspects I had so far not been aware of.

Nevertheless, there are still a few minor aspects that should be corrected/improved:

 Point 1: English language and style are fine/minor spell check required  â€¨

 Response 1: The manuscript has been revised by our department’s scientific editor.

 Point 2: It would be helpful to explain, how the different generations of EGFR inhibitors differ from each other (reversible, irreversible & targeting exclusively T790M mutants).

 Response 2: We added a paragraph in section 1 to describe how different generations of EGFR inhibitors differ from each other. Figure1 was also revised to clarify this point.

Point 3: Figure 4 suggests that EGF could bind to ERBB3 and ERBB4 – which is of course incorrect. This aspect should be corrected.

 Response 3: We corrected the figure 4 as suggested.

Point 4: An aspect worthwhile discussing in this context is the role Immune Check-Point Inhibitors play in reinvigorating tumor-specific immune responses and the effects EGFR inhibitors have on the immune system.

 Response 4:

We added two paragraphs in section 6.1 and 6.2 to discuss this issue in more depth, as suggested. We also added references 129-131 & 134.

Reviewer 3 Report

The central premise of this review is that non-small cell lung cancer (NSCLC) patients that are treated with EGFR-TKI and become resistant could be treated with YAP inhibitors. This is a timely and important review and will benefit those studying the pharmacological treatment of non-small cell lung cancer. In general the content is appropriate. The tables and figures summarize the content nicely and facilitate the reader’s understanding.

 There are two issues with the manuscript that diminish enthusiasm.

1. The writing is convoluted which makes it difficult for the reader to follow. Some of the problems are related to verb-subject agreement, use of proper verbs, and the general sentence structure. As a whole, the document would benefit from re-writing by a scientific writer.

 2. I believe 6 anti PD-1/PD-L1 drugs have been approved – avelumab and cemiplimab are missing.  These should be included to make sure the review is as comprehensive as possible.

Author Response

The central premise of this review is that non-small cell lung cancer (NSCLC) patients that are treated with EGFR-TKI and become resistant could be treated with YAP inhibitors. This is a timely and important review and will benefit those studying the pharmacological treatment of non-small cell lung cancer. In general the content is appropriate. The tables and figures summarize the content nicely and facilitate the reader’s understanding.

Point 1: The writing is convoluted which makes it difficult for the reader to follow. Some of the problems are related to verb-subject agreement, use of proper verbs, and the general sentence structure. As a whole, the document would benefit from re-writing by a scientific writer.

 Response:

The manuscript has been revised by our department’s scientific editor.

Point 2: I believe 6 anti PD-1/PD-L1 drugs have been approved – avelumab and cemiplimab are missing. These should be included to make sure the review is as comprehensive as possible.

 Response 2: We added a paragraph in section 2 and references 56-59 to discuss the anti PD-1/PD-L1 ICIs avelumab and cemiplimab as suggested. These 2 drugs are also now included in revised figure 2.

Round 2

Reviewer 3 Report

Overall, this revised manuscript reads much better and the changes in the writing make the author's point clearer.  However, it is recommended that the final article be careful proofread prior to publication.

Minor changes:

1. The sentence that reads:

“Several studies found that activation of YAP enhances the downstream gene expression of EGFs such as Amphiregulin (AREG) and Neuregulin 1 (NRG-1), and ERBB3 and ERBB4, to form an autocrine loop and reinforce the MAPK signaling pathway to induce cancer progression and drug resistance [84- 89].

Should be changed to:  

“Several studies found that activation of YAP enhances the downstream gene expression of EGFR ligands…..

2. There are still some cases of poor editing and proofreading that should be resolved before publication. For instance, inconsistent capitalization of YAP, and leaving in a note to the writer (e.g. [cite reference #).

Author Response

Point 1: 1. The sentence that reads:

“Several studies found that activation of YAP enhances the downstream gene expression of EGFs such as Amphiregulin (AREG) and Neuregulin 1 (NRG-1), and ERBB3 and ERBB4, to form an autocrine loop and reinforce the MAPK signaling pathway to induce cancer progression and drug resistance [84- 89].

Should be changed to:  

“Several studies found that activation of YAP enhances the downstream gene expression of EGFR ligands…..

 Response:

We corrected it as suggested.

Point 2: There are still some cases of poor editing and proofreading that should be resolved before publication. For instance, inconsistent capitalization of YAP, and leaving in a note to the writer (e.g. [cite reference #).

Response:

We corrected it as suggested.

The corrections are list as below:

Page 9, line 30 & line 43 & line 46; page 10, line 46; page 12, line 30
